# Estimating Markov Chain Transition Probabilities for Steady Aging Models from $n$-step Data

## Abstract

Usage-dependent aging processes of products such as batteries have become increasingly important. However, in general, revealing aging dynamics of discrete time Markov Chains from sparse $n$-step data is challenging as multi-step transition probabilities become complex and make it impossible to efficiently optimize the likelihood function for larger $n$. In this paper, we consider classes of steady-aging processes, which can be characterized by band matrices. Based on novel explicit diagonalizations of such matrices, we are able to optimally solve the likelihood in an efficient manner. In contrast to existing benchmark approaches our approach scales well and remains applicable for large $n$. Further, our evaluations for synthetic as well as real-world data verify a high estimation accuracy (about 98%) with comparably few data (about $10\times$ more observations as states).

## 1 Introduction

As the world shifts towards circular economies (Stahel, 2016; Stahel & MacArthur, 2019) the lifecycle management of products and resources has become an important topic. Aging products such as batteries are especially relevant, given their growing use in electric vehicles and renewable energy storage. Hence, for both businesses and consumers, it is crucial to understand usage-dependent aging processes and to economically plan how to use and when to replace products. In an era where end-of-life scenario management has gained greater attention, this is particularly helpful for industries dealing with high-value, high-waste products.

Understanding and quantifying product aging is crucial for economic decision-making. However, aging processes for products such as batteries are typically complex and not easy to reveal (Xiong et al., 2020; Han et al., 2024). Although general-purpose learning models are applicable, they typically require a large amount of training data. More specific approaches, like physics-inspired AI models (e.g. Kim et al. (2024); Zhang et al. (2025)), are promising but remain domain-dependent and often complex, as aging processes are typically characterized by coupled PDE models. Note, the estimation problem gets even more complex if the aging of a product also depends on a user's using patterns. A further critical aspect is whether or not a current state (e.g. the state of health (SoH) of a battery) is (i) continuously tractable and (ii) only observable in distribution.

Our goal is to provide a simple and general applicable methodological framework to reveal a product's aging dynamics from few observations without domain knowledge. Further, we seek to exploit a product's aging dynamics for optimized usage and replacement. Our contributions are:

- We estimate usage-dependent aging effects of products within a general discrete time Markov chain-based framework of bi-diagonal type using heterogeneous $n$-step transition data.
- In our approach, we propose novel explicit diagonalizations of bidiagonal matrices, which allows us to quantify aging effects from sparse single $n$-step data observations. Moreover, our fitted models remain interpretable and explainable.
- We evaluate examples for different synthetic as well as real-world battery data and compare our approach against standard solver-based approaches as well as established numerical baseline heuristics. Here, we clearly outperform other approaches in both accuracy and scalability, particularly for large $n$.

This paper is organized as follows. Section 2 discusses related works in the areas of aging diagnosis and lifecycle management. Section 3 describes the model and presents our approach to estimate

aging dynamics. In Section 4, we evaluate the accuracy of our estimation approach. In Section 5, we summarizes main results and discusses potential extensions of the model.

## 2 Related Work

**Estimation of Transition Probabilities in Discrete Markov Chains**    There is rich literature on estimation of the dynamics in Markov chains. For discrete time Markov models the most common approach is to identify the transition matrix $P$ by maximizing the log-likelihood function. For 1-step data this is a well-researched problem and various extensions exist, see, e.g. Li et al. (2018).

The case of estimating dynamics for $n$-step data is less well researched as optimizing the likelihood function becomes challenging as the entries of $n$-step transitions $P^n$ become numerically complex for larger problems, cf. the number of states. If the likelihood cannot be directly optimized numerical heuristics such as the EM algorithm (Dempster et al., 1977) are used to approximate the likelihood solution, see, e.g. Craig & Sendi (2002). However, it is also known that the EM algorithm has weaknesses if $n$ is large and the number of observations is small compared to the number of variables.

**Estimating States of Batteries and Aging Diagnosis**    The estimation of aging dynamics is an active field of research. Particularly, research about models to describe the aging of batteries is a hot topic, as recent surveys show, see Vermeer et al. (2022); Li et al. (2023); Ali et al. (2023); Xiao et al. (2023); Zeng & Liu (2023); Wang et al. (2025). A prominent line of research seeks to exploit physical rules to model charging and discharging processes of specific batteries. Albeit, the calibration of known physical dynamics is often complex and requires numerical approximations of systems of PDEs (Cordoba-Arenas et al., 2015; Chun et al., 2022; Wu et al., 2024; Zhang et al., 2024). A further major challenge in addition to calendrical aging is that the SoH of a battery can be a hidden variable, which is only observable in distribution.

More general approaches to describe aging processes using, e.g., estimated Markov chain models are less often used but also established, see Dai et al. (2019); Zhu et al. (2023); Kim et al. (2024). However, in most works, the estimation of aging dynamics requires huge amounts of data or does not support usage dependencies. Further, existing models are different from our approach, which is based on band matrices and relies on single pairs of pre- and post-state observations only.

**Aging Products and Lifecycle Management**    With the rise of circular economy models, lifecycle management research became a popular field of research, cf. Guo et al. (2021); Corno & Pozzato (2020). Besides exact methods like linear programming (LP) or dynamic programming (DP) (Weitzel & Glock, 2018), also various heuristic methods or meta-heuristics are used (Hossain et al., 2019), see, e.g., Collath et al. (2022) and references cited therein. However, existing models use only deterministic dynamics or assume highly stylized dynamics (cf. Jafari et al. (2018); Figge et al. (2022)) limiting the use of data-driven estimations within these models. In addition, most models require the current state to be deterministic and observable.

**Targeted Research Gap**    Overall, the estimation of stochastic aging dynamics for data-driven lifecycle management is still understudied. To the best of our knowledge, there is no existing estimation approach that (i) works for large $n$-step intervals, (ii) requires only sparse data, and (iii) works for observable as well as partially observable product states, that are just given in distribution. Our aim is to close that gap.

## 3 Approach to Estimate Aging Dynamics

In Section 3.1, we introduce our notation and describe the considered model for aging products. In Section 3.2, we present the problem formulation. In Section 3.3, we propose our solution approach to estimate transition dynamics from few data. Further, we summarize consequent analyses (Section 3.4) as well as generalizations and extensions of the model (Section 3.5).

### 3.1 Model Description and Data

Our goal is to estimate the stochastic aging of a certain type of product. The underlying (unknown) aging process can depend on the state of the product and its usage. We assume that we have data about

the state of the product before and after a certain episode of time as well as the type of usage that was on average applied. We consider $T$ discrete potential states, i.e., the set of states is $s \in \{1, ..., T\}$. Further, we consider $A$ different potential usage styles, i.e., the set $a \in \{1, ..., A\}$. The stochastic aging effects are described by transition probabilities, i.e., state- and usage-dependent probabilities that a product in state $s$ transitions to state $s'$ after a certain amount of time. We want to estimate such aging probabilities from given data. Let us assume that our data set consists of $K$ observations.

**Definition 1** *The observational data can be summarized as follows:*

- $x_1(k)$ *the pre-state (at begin of usage) of observation $k$*
- $x_2(k)$ *the (average) usage level (within usage) of observation $k$*
- $y(k)$ *the post-state $y$ (at end of usage) of observation $k$*
- $n(k)$ *the length (number of steps) of the usage period of observation $k$*

## 3.2 PROBLEM DESCRIPTION AND MARKOV CHAINS MODELS FOR STEADY AGING DYNAMICS

**General Aging**    Stochastic aging process for one period of time can be described as a Markovian transition probability matrix $P$ of size $T \times T$, which may depend on the usage type $a = 1, ..., A$. In the general case – which is less in focus – from each state $i$ we may reach a state $j$ within one period with a certain probability, i.e. we consider $T - 1$ probabilities for all $T$ states ($T^2 - T$ variables for all usages $a$). Note, based on a 1-step transition matrix $P$ various associated $n$-step transition probability matrices can be expressed as, $a = 1, ..., A$, $i, j = 1, ..., T$, $n = 1, 2, ...$,

$$P^n = P_{i,j}^n(a). \tag{1}$$

**Steady Aging**    When considering short periods of time and steady aging processes (without shocks), the matrix $P$ can become a *band matrix*. Here, (i) as long as periods are short, the new state of an aging product will be close to the old state, (ii) the state does not improve, and (iii) there is a terminal state $T$. This special class of aging dynamics is supported by real-world data for batteries, see Appendix A. Hence, we focus on band matrices $\bar{P}$ characterized by $T - 1$ transition probabilities $\vec{p}(a) = (p_1(a), ..., p_{T-1}(a))$, $a = 1, ..., A$, ($T - 1$ variables for all usages $a$), i.e., we use

$$\bar{P} = \bar{P}_{i,j}(\vec{p}(a)) = \begin{cases} p_i(a) & i = j < T \\ 1 - p_i(a) & i = j - 1 < T \\ 1_{\{i=j=T\}} & else, \end{cases} \tag{2}$$

where $1_{\{.\}}$ is the indicator function (and rows add up to 1). Then, based on $\bar{P}$ all $n$-step transition probabilities are given by, $a = 1, ..., A$, $i, j = 1, ..., T$, $n = 1, 2, ...$,

$$\bar{P}^n = \bar{P}_{i,j}^n(\vec{p}(a)). \tag{3}$$

Finally, our goal is to reveal usage-dependent transition probabilities $P$ from given data (see Section 3.1). In particular, we focus on the steady aging case, i.e. $\bar{P}(\vec{p}(a))$, by efficiently determining $(T-1)$-dimensional vectors $\vec{p}(a) = (p_1(a), p_2(a), ..., p_{T-1}(a))$ for different actions $a = 1, ..., A$.

## 3.3 SOLUTION APPROACH AND DIAGONALIZATION TECHNIQUES FOR STEADY AGING CASES

**Objective**    The natural approach to estimate $\bar{P}$, is to maximize the log-likelihood, i.e.

$$\max_{\vec{p}(a) \in (0,1)^{T-1}, a=1,...,A} \sum_{k=1,...,K} \log \left( \bar{P}_{x_1(k),y(k)}^{n(k)} (\vec{p}(x_2(k))) \right). \tag{4}$$

To avoid numerical problems with the log function close to 0, we use auxiliary variables $\beta_i(a) \in \mathbb{R}$, $i = 1, ..., T - 1$, $a = 1, ..., A$, to express the probabilities $p_i(a) \in (0, 1)$ via the relation $p_i(a) = \exp(\beta_i(a))/\exp(1 + \beta_i(a)) \in (0, 1)$, $i = 1, ..., T - 1$, $a = 1, ..., A$.

**Limitations of Standard Approaches**    To address *general* aging dynamics, in equation 4, we can also use matrices $P$ (with up to $T^2 - T$ variables) instead of band matrices $\bar{P}$. In both cases, if $n(k)$ is not small, due to the associated number of matrix multiplications, the problem is, in general, not tractable by standard solvers. Alternatively, one can define the $n$-step probabilities $P^n$ as co-variables based on $P$. This, however, requires more memory to store all the (non-sparse) tables of co-variables $P^n$, $n = 1, ..., N$ (dimensions are in general $T \times T \times A \times N$ and about $T \times T/2 \times A \times N$ if $P$ is a band matrix). Consequently, the tractability of solving objective 4 is known to be limited to smaller problems or iterative approximations, e.g., using expectation maximization have to be used.

**Our Diagonalization Approach** We focus on bi-diagonal matrices, cf. steady aging models, particularly, for the challenging case of large and heterogeneous $n$. We seek to circumvent the known limitations of standard approaches by using diagonalizations of $\bar{P}(\vec{p}(a))$. We recall that numerical standard diagonalizations for fixed number matrices are *not* applicable in this case, as for all $a$ we need diagonalizations of $\bar{P}(\vec{p})$ as *explicit function* of the $T-1$ variables $\vec{p}$. While such formulas are in general not tractable, we propose such explicit solution formulas for general given bi-diagonal matrices $\bar{P}(\vec{p})$, cf. equation 2.

**Theorem 1** *For positive integer $T$ the $T \times T$ band matrix $\bar{P}(\vec{p})$ of the form equation 2, can be expressed as*

$$\bar{P}(\vec{p}) = U(\vec{p}) \cdot D(\vec{p}) \cdot V(\vec{p}), \tag{5}$$

*where, $i, j = 1, ..., T$,*

$$D_{i,j}(\vec{p}) = \begin{cases} 1 & i = j = 1 \\ p_{i-1} & i = j > 1 \\ 0 & else \end{cases} \tag{6}$$

$$U_{i,j}(\vec{p}) = \begin{cases} 1 & j = 1 \vee j = i+1 \\ 0 & j > 1 \wedge i \geq j \\ \frac{\lambda(i,j-2)}{\kappa(i,j-1)} \cdot (-1)^{i+j-1} & else \end{cases} \tag{7}$$

$$V_{i,j}(\vec{p}) = \begin{cases} 1_{\{j=T\}} & i = 1 \\ 1 & i = j+1 \\ 0 & i > j+1 \\ -1 & i = j = T \\ \frac{\lambda(i,T-1)}{\theta(i-1,T-1)} \cdot (-1)^{i+j-1} & i < j = T \\ \frac{\lambda(i-1,j-1)}{\theta(i-1,j)} \cdot (-1)^{i+j-1} & else \end{cases} \tag{8}$$

*with $\lambda(i,j) := \prod\limits_{k=i,...,j} (p_k - 1)$, $\kappa(i,j) := \prod\limits_{k=i,...,j-1} (p_j - p_k)$, $\theta(i,j) := \prod\limits_{k=i+1,...,j} (p_i - p_k)$.*

*Proof.* From simple algebra follows that (i) $U$ is the inverse of $V$, i.e., $U(\vec{p}) \cdot V(\vec{p}) = 1^{T \times T}$ yields the identity matrix, and (ii) we indeed have $\bar{P}(\vec{p}) = U(\vec{p}) \cdot D(\vec{p}) \cdot V(\vec{p})$ for all rows $i = 1, ..., T$ and columns $j = 1, ..., T$. For further details, see Appendix B. $\square$

The closed-form expressions of our Theorem 1 can be used to extremely simplify equation 9 to avoid both previous limitations, i.e., computational effort *and* required memory for co-variables, as we obtain all single values $\bar{P}^{n(k)}_{i,j}$, cf. equation 3, by $\bar{P}^{n(k)}_{i,j}(\vec{p}(a)) = \sum_{t=1,...,T} U_{i,t}(\vec{p}(a)) \cdot D^{n(k)}_{t,t}(\vec{p}(a)) \cdot V_{t,j}(\vec{p}(a))$, $i, j = 1, ..., T$, where the matrix $D^{n(k)}(\vec{p}(a))$, cf. equation 6, has the diagonal elements $D^{n(k)}_{1,1}(\vec{p}(a)) = 1$ and $D^{n(k)}_{i,i}(\vec{p}(a)) = p_{i-1}(a)^{n(k)}$, $i = 2, ..., T$, for all $a = 1, ..., A$, $k = 1, ..., K$.

## 3.4 FUTURE STATE PREDICTION, DURATIONS, AND EXPECTED LIFETIME

Being able to accurately estimate transition probabilities has far reaching implications and can be of high interest for both consumers (users of electrical cars/batteries) and industry (e.g. car-sharing firms, battery manufacturers, short/long-term battery renting/leasing). The prediction of future states and expected lifetimes can be easily derived based on $\bar{P}$.

**State Evolution Prediction** Given (estimated) transition probabilities $\vec{p}(a)$ predictions regarding future state evolutions require the current state (not a sequence of states as in various time series models). For any state-dependent action (policy) $a(s)$ and a current state $s$ the associated distribution of future states in $n$ periods is given by $\bar{P}^n(\vec{p}(a(s)))$, $n = 1, 2, ....$

**Durations and Expected Lifetime** By $D_s$ we denote the duration the product remains in a state $s$, $s = 1, ..., T-1$, under policy $a(s)$. The probability that $D_s$ equals $n$ periods is (geometric distribution), $n \geq 1$,

$$P(D_s = n) = (1 - p_s(a(s))) \cdot p_s(a(s))^n.$$

Hence, the expected duration in state $s = 1, ..., T - 1$ for action $a$ is $E(D_s) = 1/\left(1 - p_s(a(s))\right) - 1$. Based on that, we can determine the expected lifetime $(L)$, i.e., the average time of the aging process from state 1 to the final state $T$. The expected cycle length amounts to $E(L) = \sum_{s=1,...,T-1} E(D_s)$.

### 3.5 Model Generalizations and Extensions

In the following, we propose generalization and extensions of our model framework, which remain compatible with the diagonalization approach.

**Generalization: Input Data for States and Actions as Belief Distribution**  Our model can also be used with more general input data. This can be the case if, e.g. the state-of-health of a battery can only be measured as a probability distribution.

**Definition 2** *We consider that observational data can be given as belief distributions:*

- $\vec{x}_1(k) = (x_1(k, 1), ..., x_1(k, T))$ *the prob. distribution for pre-state $x_1$ of observation $k$*
- $\vec{x}_2(k) = (x_2(k, 1), ..., x_2(k, A))$ *the prob. distribution for the usage $x_2$ of observation $k$*
- $\vec{y}(k) = (y(k, 1), ..., y(k, T))$ *the prob. distribution for the post-state $y$ of observation $k$*

Note, in case the data observations' pre-states ($x_1$), post-states ($y$), and usage levels ($x_2$) are *randomized* (i.e., given as distributions) – as a generalization of the objective equation 4 – we can estimate all $\vec{p}(a)$, $a = 1, ..., A$, by minimizing the Kullback-Leibler (KL) divergence between the distributions of states $\vec{y}$ and *fitted* states $\vec{\hat{y}}$, i.e., $KL(\vec{y}(k)||\vec{\hat{y}}(k)) = \sum_j y(k, j) \cdot \log\left(y(k, j)/\hat{y}(k, j; \vec{p}(a))\right)$, over $\vec{p}(a)$, where, given $\vec{x}_1(k)$, $\vec{x}_2(k)$, and $n(k)$, the distribution of fitted post states is $\hat{y}(k, j; \vec{p}(a)) = \sum_{a=1,...,A} x_2(k, a) \cdot \sum_{i=1,...,T} x_1(k, i) \cdot \bar{P}_{i,j}^{n(k)}(\vec{p}(a))$, $j = 1, ..., T$, cf. equation 3. Hence, given $K$ data observations, we estimate $\vec{p}(a)$, $a = 1, ..., A$, via minimizing the sum of the KL divergences, i.e.,

$$\min_{\vec{p}(a) \in (0,1)^{T-1}, a=1,...,A} \sum_{k=1,...,K} \sum_{a=1,...,A} x_2(k, a) \cdot \sum_{j=1,...,T} y(k, j)$$
$$\cdot \left(\log\left(y(k, j)\right) - \log\left(\sum_{i=1,...,T} x_1(k, i) \cdot \bar{P}_{i,j}^{n(k)}(\vec{p}(a))\right)\right). \tag{9}$$

**Feature-based Variable Reduction**  As our model is flexible it can be even further modified by expressing the $(T - 1) \cdot A$ variables $\beta_i(a)$ (and thus associated $p_i(a)$), e.g., using linear basis function models to reduce the number of variable to $B$ variables only ($B << T - 1$). Using some features $f_b$, $b = 0, 1, ..., B - 1$, we define $\beta_i(a) = \alpha_0 + \alpha_1 \cdot f_1(i, a) + ... + \alpha_{B-1} \cdot f_{B-1}(i, a)$, $i = 1, ..., T - 1$, and optimize for the new variables $\alpha_b \in \mathbb{R}$, $b = 0, ..., B - 1$, only.

## 4 Evaluation

### 4.1 Baseline Approaches

To estimated dynamics from heterogeneous $n$-step data, we compare our approach against the following established standard benchmark approaches:

- General Standard Solver (GSS): Solves the log-likelihood for $P$ with $T \times (T - 1)$ variables
- Bi-diagonal Standard Solver (BSS): Solves the log-likelihood for $\bar{P}$ with $(T - 1)$ variables
- Expectation Maximization (EM): Approximates the log-likelihood iteratively (for $\bar{P}$ with $(T - 1)$ variables)
- Bi-diagonal Diagonalization Solver (BDS, **ours**): Solves the log-likelihood for $\bar{P}$ with $(T - 1)$ variables, cf. objective equation 4 (or 9) using the representations of Theorem 1.

GSS serves as a benchmark for problems where the data is not exactly of bi-diagonal form (cf. real-world data). GSS does not scale at all due to the number of variables determining $P$ and auxiliary variables for $P^n$. BSS also does not scale due to the number of variables determining $P$ and auxiliary variables for $P^n$. EM scales better than GSS and BSS but only approximated the log-likelihood and might find only local optima. Next, we will evaluate how BDS relates to the other baselines.

## 4.2 APPLICABILITY OF OUR APPROACH COMPARED TO STANDARD METHODS

**Example 1** *(Applicability and Runtimes)* *To study scalability we considered problems with increasing $N$ (for setups with $T = 50$, $A = 5$, $K = 2\,000$, and $P$ a band matrix). We used an AMPL implementation on a notebook with Intel(R) Core(TM) i7-3630QM CPU 2.40GHz, 20GB RAM as well as the off-the-shelf solver MINOS (Murtagh & Saunders, 1978) (Version 5.5).*

Table 1: Scalability Comparison for different $N$: BDS compared to other Baselines (for problems with $T = 50$, $A = 5$, $K = 2\,000$, and $P = \bar{P}$ a band matrix); runtimes in seconds for Example 1.

| $N$ | MEAN SOLVE TIME BSS | MEAN RUNTIME EM | MEAN SOLVE TIME BDS (OURS) |
|---|---|---|---|
| 1 | 0.3 s | 0.3 s | 0.3 s |
| 5 | 0.6 s | 3.5 s | 0.4 s |
| 10 | 1.8 s | 12 s | 0.9 s |
| 15 | OOM | 25 s | 1.5 s |
| 50 | OOM | 90 s | 2.8 s |
| 100 | OOM | DNF | 3.3 s |
| 500 | OOM | DNF | 4.6 s |
| 1 000 | OOM | DNF | 5.8 s |

For small ($N \leq 5$) BSS and our BDS approach perform equally, see Table 1. EM requires more time to converge. For $N \geq 15$ the standard BSS approach runs out-of-memory (OOM) and is not applicable anymore. For larger $N$ EM did not finish (DNF) in under 100 s. This verifies that our diagonalization approach is faster than BSS and EM – and most importantly, the only one that is applicable for larger $N$, which is crucial for applications with occasional and delayed data.

Instead, solving the model with BDS remains tractable even if $N = 100$. Moreover, the runtime of the diagonalization model even scales well in $N$. This is because BDS allows to explicitly express $n$-step transition probabilities by combining to key elements: (i) a tractable form and (ii) keeping large tables of co-variables is not required. The number of co-variables in the (non-sparse) tables does not depend on $N$ (i.e. dimensions are $T \times T \times A \times 5$, where the number 5 is due to $U$, $V$, $\lambda$, $\kappa$, and $\theta$). Hence, the diagonalization pays off if $N > 5$.

## 4.3 RESULTS FOR SYNTHETIC AND REAL-WORLD DATA

To test the accuracy of our methodology, we consider an exemplary aging matrix $\bar{P}$ (ground truth) to be revealed. We first consider the case of observable states and actions, cf. Definition 1. and solve equation 4 using BDS, which dominates BSS in solve time and EM in both runtime and accuracy. We again use the solver MINOS (Murtagh & Saunders, 1978) (Version 5.5) via AMPL on a consumer notebook with Intel(R) Core(TM) i7-3630QM CPU 2.40GHz, 20GB RAM.

**Example 2** *(Accuracy against Synthetic Ground Truth and Impact of Model Parameters)* *We consider the Base Case: $A = 1$, $T = 20$, $N = 20$, $G = 1/N$, with observable pre-states $x_1$ and actions $x_2$ using discrete uniform random realizations, i.e. $U(1, T - 1)$ and $U(1, A)$; for the interval lengths $n$, we use $U(G \cdot N, N)$. Further, for reproducibility, we consider the (true) exemplary probabilities $p_i(a) = 1 - \frac{1}{2} \cdot \frac{a+i}{A+T} \cdot \frac{1}{N^{0.5}}$ and simulate corresponding post-states $y$ for $K = 1\,000$ observations. We use 10 repeated runs to evaluate average results and their standard deviations.*

Note, in our example, the aging probabilities $\vec{p}$ are state- and action-dependent. The probabilities to leave the current state are increasing in the usage level $a$ (reflecting usage intensity) and increasing with older states $s$ compared to the initial (new) state $s = 1$, cf. accelerated aging Li et al. (2023). Further, we account for the considered maximum interval length of $N$ such that with increasing $N$ the probabilities to remain in the current state are closer to 1; otherwise the probabilities to remain in a state different from $T$ would be quickly going to zero with increasing numbers of $N$.

Table 2 shows that with more states $T$ the results are less accurate (MAPE = Mean Absolute Percentage Error for estimated $\hat{p}_i(a)$ vs. true $p_i(a)$, $i = 1, ..., T - 1$, $a = 1, ..., A$; MAE = Mean Absolute Error). This is reasonable as more variables have to be estimated with the same amount of data. Table 3 shows that with higher $N$ the regression results improve. This is due to the fact that $\vec{p}$ are increasing in $N$ and thus less diverse.

Table 2: Impact of the number of states $T$; Example 2.

| $T$ | MAPE | MAE | TIME IN S |
|---|---|---|---|
| 5 | $0.004\pm 0.001$ | $0.004\pm 0.001$ | $0.02\pm 0.005$ |
| 10 | $0.006\pm 0.002$ | $0.005\pm 0.001$ | $0.04\pm 0.004$ |
| 20 | $0.009\pm 0.001$ | $0.008\pm 0.001$ | $0.10\pm 0.009$ |
| 30 | $0.011\pm 0.001$ | $0.011\pm 0.001$ | $0.12\pm 0.020$ |
| 50 | $0.014\pm 0.002$ | $0.013\pm 0.002$ | $0.17\pm 0.018$ |
| 70 | $0.016\pm 0.001$ | $0.015\pm 0.001$ | $0.23\pm 0.013$ |
| 100 | $0.020\pm 0.001$ | $0.019\pm 0.001$ | $0.33\pm 0.015$ |

Table 3: Impact of the maximum number of steps $N$; Example 2.

| $N$ | MAPE | MAE | TIME IN S |
|---|---|---|---|
| 1 | $0.060\pm 0.013$ | $0.042\pm 0.009$ | $0.03\pm 0.003$ |
| 5 | $0.024\pm 0.003$ | $0.020\pm 0.002$ | $0.04\pm 0.006$ |
| 10 | $0.016\pm 0.002$ | $0.014\pm 0.002$ | $0.07\pm 0.005$ |
| 20 | $0.009\pm 0.002$ | $0.008\pm 0.002$ | $0.10\pm 0.010$ |
| 50 | $0.005\pm 0.001$ | $0.005\pm 0.001$ | $0.15\pm 0.011$ |
| 100 | $0.003\pm 0.001$ | $0.003\pm 0.001$ | $0.19\pm 0.012$ |
| 200 | $0.002\pm 0.001$ | $0.002\pm 0.001$ | $0.25\pm 0.011$ |
| 500 | $0.001\pm 0.001$ | $0.002\pm 0.001$ | $0.37\pm 0.025$ |
| 1 000 | $0.001\pm 0.001$ | $0.001\pm 0.001$ | $0.49\pm 0.033$ |

In the Appendix E, we show further tables studying the impact of $K$, $G$, and $A$, see Table 9 - 11. Table 9 shows that with more observations $K$ the regression results get better. Naturally, this can be expected. Instead, parameter $G \in [0, 1]$ affects the usage lengths $n(k)$ to be *at least* $G \cdot N$. Table 10 shows that with longer interval lengths, the regression results are not affected. This is a nice result. Note, with short interval lengths, e.g., $n = 1$, the aging matrix, cf. $\vec{p}$, can be directly estimated. Nevertheless, the results show that longer interval lengths are sufficient to estimate $\vec{p}$, which highlights the general applicability of the model refraining from the need of having data for short intervals. Table 11 shows that results in $A$ remain accurate although $K$ is fixed.

**Example 3** *(**Real-World Battery Data**) We use $A = 1$, $T = 10$, $N = 20$, and data with observable states, cf. $x_1$ and $y$. For reproducibility, see Appendix A. The results for using (i) a general aging matrix $P$ and (ii) a band matrix $\bar{P}$ almost coincide; they are summarized in Tables 4 and 5.*

In this example, we tested our BDS model using a real-world dataset. For comparison, we used the general model, cf. GSS, with (i) full entries $P$ and (ii) our band matrix model, cf. BDS. Table 4 shows the results for the general full model. We find that the entries of the fitted full matrix are similar to a band matrix, which verifies that our band matrix assumption holds in this real-world application.

Table 4: Results when using a general aging matrix $P$: Fitted values for 1-step transition probabilities $P$ with full $T^2 - T$ variables $p_{i,j}$, $i = 1, ..., T-1$, $j = 1, ..., T$ ($K = 3\,150$ observations, $T = 10$, $N = 20$ with $n(k)$ from 1 to 20, general model solved via GSS; Example 3.

| $i\backslash j$ | 1 | 2 | 3 | 4 | 5 | 6 | 7 | 8 | 9 | 10 |
|---|---|---|---|---|---|---|---|---|---|---|
| 1 | 0.780 | 0.220 | 0.000 | 0.000 | 0.000 | 0.000 | 0.000 | 0.000 | 0.000 | 0.000 |
| 2 | 0.000 | 0.909 | 0.091 | 0.000 | 0.000 | 0.000 | 0.000 | 0.000 | 0.000 | 0.000 |
| 3 | 0.000 | 0.020 | 0.903 | 0.076 | 0.000 | 0.000 | 0.000 | 0.000 | 0.000 | 0.000 |
| 4 | 0.000 | 0.000 | 0.014 | 0.919 | 0.067 | 0.000 | 0.000 | 0.000 | 0.000 | 0.000 |
| 5 | 0.000 | 0.000 | 0.000 | 0.000 | 0.893 | 0.107 | 0.000 | 0.000 | 0.000 | 0.000 |
| 6 | 0.000 | 0.000 | 0.000 | 0.000 | 0.000 | 0.912 | 0.088 | 0.000 | 0.000 | 0.000 |
| 7 | 0.000 | 0.000 | 0.000 | 0.000 | 0.000 | 0.019 | 0.942 | 0.039 | 0.000 | 0.000 |
| 8 | 0.000 | 0.000 | 0.000 | 0.000 | 0.000 | 0.000 | 0.003 | 0.961 | 0.036 | 0.000 |
| 9 | 0.000 | 0.000 | 0.000 | 0.000 | 0.000 | 0.000 | 0.000 | 0.000 | 0.959 | 0.041 |
| 10 | 0.000 | 0.000 | 0.000 | 0.000 | 0.000 | 0.000 | 0.000 | 0.000 | 0.000 | 1.000 |

Table 5: Results when using a band matrix $\bar{P}$ (cf. BDS): Fitted values for bi-diagonal 1-step transition probabilities $\bar{P}$ with only $T-1$ variables $p_{i.i}$, $i = 1, ..., T-1$ ($K = 3\,150$ data observations, $T = 10$, $N = 20$ with $n(k)$ from 1 to 20, band matrix model solved via BDS; Example 3.

| $i\backslash j$ | 1 | 2 | 3 | 4 | 5 | 6 | 7 | 8 | 9 | 10 |
|---|---|---|---|---|---|---|---|---|---|---|
| 1 | 0.780 | 0.220 | 0 | 0 | 0 | 0 | 0 | 0 | 0 | 0 |
| 2 | 0 | 0.921 | 0.079 | 0 | 0 | 0 | 0 | 0 | 0 | 0 |
| 3 | 0 | 0 | 0.930 | 0.070 | 0 | 0 | 0 | 0 | 0 | 0 |
| 4 | 0 | 0 | 0 | 0.931 | 0.069 | 0 | 0 | 0 | 0 | 0 |
| 5 | 0 | 0 | 0 | 0 | 0.892 | 0.108 | 0 | 0 | 0 | 0 |
| 6 | 0 | 0 | 0 | 0 | 0 | 0.924 | 0.076 | 0 | 0 | 0 |
| 7 | 0 | 0 | 0 | 0 | 0 | 0 | 0.961 | 0.039 | 0 | 0 |
| 8 | 0 | 0 | 0 | 0 | 0 | 0 | 0 | 0.964 | 0.036 | 0 |
| 9 | 0 | 0 | 0 | 0 | 0 | 0 | 0 | 0 | 0.959 | 0.041 |
| 10 | 0 | 0 | 0 | 0 | 0 | 0 | 0 | 0 | 0 | 1.000 |

Table 5 shows the results for our BDS model. As expected, results are similar to those of the general model and can be computed more efficiently.

## 4.4 EXTENSIONS: DATA WITH BELIEF DISTRIBUTIONS AND VARIABLE REDUCTION

Next, we consider the more general case when data is not directly observable and consists only of belief distributions for $x_1$, $x_2$, and $y$ (Definition 2). The estimation is now solved via KL-objective 9.

**Example 4** *(Generalized Input Data) We consider the Base Case of Example 2 (i.e., $A = 1$, $T = 20$, $N = 20$, $G = 1/N$) but now we use distributions for given states and actions, cf. Definition 2, i.e., we use $\vec{x}_1$, $\vec{x}_2$, and $\vec{y}$ as data, which are distributed uniformly with $W = 1$ (i.e. $(1/3, 1/3, 1/3)$ for differences $\{-1, 0, 1\}$) around random 1-dim. realizations as used in Example 2. Further, we let $K = T \cdot A \cdot 50$ to account for different numbers of variables, cf. $p_i(a)$. We evaluate 10 runs.*

For the case where data for actions, pre- and post-states is given by belief distributions, Table 6 summarizes results of different $T$ for the setup of Example 4. We observe that the accuracy remains accurate even for larger $T$.

Table 6: Impact of the number of states $T$; Example 4.

| $T$ | MAPE | MAE | TIME IN S |
|---|---|---|---|
| 5 | $0.010\pm 0.003$ | $0.009\pm 0.003$ | $0.03\pm 0.007$ |
| 10 | $0.010\pm 0.002$ | $0.009\pm 0.002$ | $0.09\pm 0.009$ |
| 20 | $0.009\pm 0.005$ | $0.008\pm 0.004$ | $0.42\pm 0.037$ |
| 30 | $0.013\pm 0.010$ | $0.012\pm 0.009$ | $1.16\pm 0.116$ |
| 50 | $0.011\pm 0.008$ | $0.010\pm 0.007$ | $2.29\pm 1.371$ |
| 70 | $0.024\pm 0.008$ | $0.023\pm 0.007$ | $5.11\pm 1.524$ |
| 100 | $0.078\pm 0.051$ | $0.073\pm 0.048$ | $8.96\pm 3.640$ |

In the Appendix F, we show further tables studying the impact of $N$, $K$, and $A$, see Tables 12, 13, and 14. The results for $N$, see Table 12, are similar to Table 3. Table 13 shows the impact of the number of observations $K$. As expected, the estimation accuracy is increasing in $K$. More importantly, we obtain that results are again already competitive for comparably few ($K = 100$) data points.

Table 14 shows the impact of the number $A$ of different usage levels in case also the usage levels are given by belief distributions. We observe that with increasing $A$ the estimation accuracy is slightly decreasing. However, the results verify that our approach for belief distributions, cf. equation 9, remains applicable even for multiple usage levels $A$. Overall, compared to Example 2, the accuracies are slightly worse, which is due to the stochasticity in the data, and particularly due to multiple usage levels $A$ where the estimation of the different $\vec{p}(a)$ are coupled within equation 9.

**Example 5** *(Variable Reduction)* *We consider the base case Example 2 with $B = 3$ features. Following Section 3.4, we define $\beta_i(a) = \alpha_0 \cdot f_0(i,a) + \alpha_1 \cdot f_1(i,a) + \alpha_2 \cdot f_2(i,a)$ and use the features $f_0(i,1) := 1$, $f_1(i,1) := i$, and $f_2(i,1) := i^{0.5}$. Further, we vary the number of states $T$.*

Table 7: Results for using linear basis function models for different numbers of states $T$; Example 5.

| $T$ | MAPE | MAE | TIME IN S |
|---|---|---|---|
| 5 | $0.003 \pm 0.001$ | $0.003 \pm 0.001$ | $0.02 \pm 0.006$ |
| 10 | $0.003 \pm 0.001$ | $0.003 \pm 0.001$ | $0.03 \pm 0.013$ |
| 20 | $0.002 \pm 0.001$ | $0.002 \pm 0.001$ | $0.03 \pm 0.015$ |
| 30 | $0.005 \pm 0.001$ | $0.004 \pm 0.001$ | $0.04 \pm 0.007$ |
| 50 | $0.004 \pm 0.000$ | $0.004 \pm 0.000$ | $0.04 \pm 0.003$ |
| 70 | $0.003 \pm 0.000$ | $0.003 \pm 0.000$ | $0.03 \pm 0.005$ |
| 100 | $0.014 \pm 0.005$ | $0.013 \pm 0.005$ | $0.04 \pm 0.008$ |

Instead of $T - 1$ variables our BDS approach now uses 3 variables only ($\alpha_k$, $k = 0, 1, 2$) to describe all $p_i$, $i = 1, ..., T - 1$. The results, see Table 7, verify that approach can be used to make the number of variables independent of $T$, which in turn, allows to significantly reduce the solve time for larger $T$. At the same time, in the considered example, we assume that the estimations remain accurate (cp. Table 2). Similarly, if the number of actions $A$ is large and actions are metric (e.g. the measure a usage intensity or a frequency) the approach can be used similarly to reduce the required number of variables by using action-dependent features.

## 5 DISCUSSION AND CONCLUSION

**Main Results**   We proposed a fairly general approach to estimate aging dynamics. The data can be incomplete and have longer intervals without observations. Besides, the model also remains applicable when pre- and post-states are only observed in distribution. Our sample efficiency is based on Theorem 1, which uses explicit diagonalization expressions and allows to target problems with $N >> T$, see Appendix C, using a fraction of variables compared to standard approaches.

Problems equation 9 and equation 4 can be numerically solved using standard continuous non-linear solvers. While standard approaches without diagonalizations are limited to small problems, our diagonalization approach allows to solve problems assuming steady aging (as in real-life battery applications, see Appendix A) with up to $N \geq 1\,000$ periods in just a few seconds, see Section 4.2.

**Assumptions and Limitations**   We make the following assumptions. First, we assume that the state (e.g. of the product) is discrete. In Example 5, it has to be metric. Second, in the current model, we use band matrices refraining from larger state changes or shocks as well as improving states. Third, we assume that the (average) usage level associated to a certain observation was constant over time.

**Future Work Directions**   The model could be further extended regarding (i) the bandwidth of the matrices, (ii) the state's dimensions, or (iii) a continuous state space. Other future research directions are to estimate aging probabilities via Bayesian approaches (Jafari et al., 2019), e.g., by exploiting conjugacy of the Beta distribution via adaptive prior/posterior updates for Bernoulli or Binomial likelihoods. Moreover, revealed usage-dependent transition probabilities can also be used for optimizing life cycle management (with product replacement), i.e., which action to use in which state, using dynamic programming techniques.

**Conclusion**   We have proposed a both general and simple model to detect usage-dependent dynamics for a special class of steady-aging products (such as batteries). Using synthetic as well as real-world battery data we have shown that the model is easy to calibrate and requires few data observations allowing for episodes without any data. While existing estimation approaches are not applicable for larger $n$-step data, our method remains applicable for large $n$ and retains fast runtimes to optimally solve the likelihood function. Further, we verify that our model provides near-optimal prediction results for scenarios with observable state data as well as probabilistic state beliefs. Hence, the proposed model is useful for a variety of applications, particularly, if state observations are sparse, costly, or occur only occasionally, cf., e.g., electrical cars/batteries, car-sharing applications, battery manufacturers, short/long-term battery renting/leasing, etc.

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

## A    REAL-WORLD BATTERY DATA

We applied our model to real-world battery data using the NASA dataset: *https://www.nasa.gov/intelligent-systems-division/discovery-and-systems-health/pcoe/pcoe-data-set-repository/* by B. Saha and K. Goebel (2007). "Battery Data Set", NASA Prognostics Data Repository, NASA Ames Research Center, Moffett Field, CA.

As an example, for battery B0006, we linearly mapped the remaining capacity with minimum (1.154 Ahr) and maximum values (2.035 Ahr) to our state representation $i = 1, ..., T$, with choosable $T$, where 1 is the best and $T$ is the worst or terminal state.

To get an overview of the entire dataset, see file meta.csv in our codebase (column "capacity", the data we use is extracted specifically from B0006.mat in the NASA dataset). See *https://anonymous.4open.science/r/ProUsAgE-5D29/README.md* for the codebase.

From the provided time series for remaining capacity we collected $K = 3150$ observation with $n$-step transitions between 1 and $N = 20$. 1-step data for single batteries typically look as follows (see Table 8 below).

Table 8: Number of transitions from state $i = x_1$ (rows) to $j = y$ (columns) for real-world data; $K = 167$ observations, $T = 10$, for 1-step intervals $n(k) = 1$, $k = 1, ..., K$ (battery B0006).

| $i\backslash j$ | 1 | 2 | 3 | 4 | 5 | 6 | 7 | 8 | 9 | 10 |
|---|---|---|---|---|---|---|---|---|---|---|
| 1 | 6 | 1 | 0 | 0 | 0 | 0 | 0 | 0 | 0 | 0 |
| 2 | 0 | 13 | 3 | 0 | 0 | 0 | 0 | 0 | 0 | 0 |
| 3 | 0 | 2 | 12 | 2 | 0 | 0 | 0 | 0 | 0 | 0 |
| 4 | 0 | 0 | 1 | 13 | 1 | 0 | 0 | 0 | 0 | 0 |
| 5 | 0 | 0 | 0 | 0 | 8 | 1 | 0 | 0 | 0 | 0 |
| 6 | 0 | 0 | 0 | 0 | 0 | 15 | 3 | 0 | 0 | 0 |
| 7 | 0 | 0 | 0 | 0 | 0 | 2 | 24 | 2 | 0 | 0 |
| 8 | 0 | 0 | 0 | 0 | 0 | 0 | 1 | 26 | 1 | 0 |
| 9 | 0 | 0 | 0 | 0 | 0 | 0 | 0 | 0 | 20 | 1 |
| 10 | 0 | 0 | 0 | 0 | 0 | 0 | 0 | 0 | 0 | 9 |

The data reflects a state-dependent aging dynamic and also includes occasionally state improvements (capacitor regeneration phenomenon). Looking at the 167 1-step transitions, we have 146 transition from $i$ to $i$, 6 transitions from $i$ to $i-1$ (state improvements), 15 transitions from $i$ to $i+1$ (state degradation), and 0 transitions from $i$ to $i+2$ (strong degradation). Hence, the aging dynamic overall fits to the bi-diagonal Markov chain model.

## B    PROOF OF THEOREM 1 (SECTION 3.3)

The formulas of interest can be written as (to assign plus/minus signs, without loss of generality, we assume $T$ even):

$$\bar{P} = \bar{P}_{i,j}(\vec{p}) = \begin{cases} p_i & i = j < T \\ 1 - p_i & i = j - 1 < T \\ 1_{\{i=j=T\}} & else, \end{cases} = \begin{pmatrix} p_1 & 1-p_1 & 0 & \cdots & \cdots & 0 \\ 0 & p_2 & 1-p_2 & 0 & \cdots & 0 \\ \vdots & 0 & \ddots & \ddots & 0 & \vdots \\ \vdots & \vdots & \ddots & \ddots & \ddots & 0 \\ \vdots & \vdots & 0 & 0 & p_{T-1} & 1-p_{T-1} \\ 0 & 0 & \cdots & \cdots & 0 & 1 \end{pmatrix}$$

$$U_{i,j}(\vec{p}) = \begin{cases} 1 & j = 1 \vee j = i+1 \\ 0 & j > 1 \wedge i \geq j \\ \frac{\lambda(i,j-2)}{\kappa(i,j-1)} \cdot (-1)^{i+j-1} & else \end{cases} = \begin{pmatrix} 1 & 1 & -\frac{\lambda(1,1)}{\kappa(1,2)} & \frac{\lambda(1,2)}{\kappa(1,3)} & \cdots & \frac{\lambda(1,T-2)}{\kappa(1,T-1)} \\ 1 & 0 & 1 & -\frac{\lambda(2,2)}{\kappa(2,3)} & \cdots & -\frac{\lambda(2,T-2)}{\kappa(2,T-1)} \\ \vdots & 0 & 0 & 1 & \ddots & \vdots \\ \vdots & \vdots & \ddots & \ddots & \ddots & \frac{\lambda(T-2,T-2)}{\kappa(T-2,T-1)} \\ \vdots & \vdots & 0 & \ddots & \ddots & 1 \\ 1 & 0 & \cdots & \cdots & \cdots & 0 \end{pmatrix}$$

$$D_{i,j}(\vec{p}) = \begin{cases} 1 & i = j = 1 \\ p_{i-1} & i = j > 1 \\ 0 & else \end{cases} = \begin{pmatrix} 1 & 0 & \cdots & \cdots & \cdots & 0 \\ 0 & p_1 & 0 & & \cdots & 0 \\ \vdots & 0 & p_2 & \ddots & 0 & \vdots \\ \vdots & \vdots & \ddots & \ddots & \ddots & \vdots \\ \vdots & \vdots & 0 & \ddots & \ddots & 0 \\ 0 & 0 & \cdots & \cdots & 0 & p_{T-1} \end{pmatrix}$$

$$V_{i,j}(\vec{p}) = \begin{cases} 1_{\{j=T\}} & i = 1 \\ 1 & i = j+1 \\ 0 & i > j+1 \\ -1 & i = j = T \\ \frac{\lambda(i,T-1)}{\theta(i-1,T-1)} \cdot (-1)^{i+j-1} & i < j = T \\ \frac{\lambda(i-1,j-1)}{\theta(i-1,j)} \cdot (-1)^{i+j-1} & else \end{cases}$$

$$= \begin{pmatrix} 0 & 0 & 0 & \cdots & \cdots & 0 & 1 \\ 1 & -\frac{\lambda(1,1)}{\theta(1,2)} & \frac{\lambda(1,2)}{\theta(1,3)} & -\frac{\lambda(1,3)}{\theta(1,4)} & \cdots & \frac{\lambda(1,T-2)}{\theta(1,T-1)} & -\frac{\lambda(2,T-1)}{\theta(1,T-1)} \\ 0 & 1 & -\frac{\lambda(2,2)}{\theta(2,3)} & \frac{\lambda(2,3)}{\theta(2,4)} & \ddots & -\frac{\lambda(2,T-2)}{\theta(2,T-1)} & \frac{\lambda(3,T-1)}{\theta(2,T-1)} \\ \vdots & \ddots & \ddots & \ddots & \ddots & \vdots & \vdots \\ \vdots & \ddots & \ddots & \ddots & \ddots & \frac{\lambda(T-3,T-2)}{\theta(T-3,T-1)} & -\frac{\lambda(T-2,T-1)}{\theta(T-3,T-1)} \\ \vdots & \ddots & 0 & \ddots & \ddots & -\frac{\lambda(T-2,T-2)}{\theta(T-2,T-1)} & \frac{\lambda(T-1,T-1)}{\theta(T-2,T-1)} \\ 0 & \cdots & \cdots & \cdots & 0 & 1 & -1 \end{pmatrix}$$

where $\lambda(i,j) := \prod_{k=i,\dots,j} (p_k - 1)$, $\kappa(i,j) := \prod_{k=i,\dots,j-1} (p_j - p_k)$, and $\theta(i,j) := \prod_{k=i+1,\dots,j} (p_i - p_k)$, $i,j = 1,\dots,T$.

Considering, for instance, for $i = j = 3$, by multiplying the $i$-th row of $U$ with the $j$-th column of $V$, we obtain $(U \cdot V)_{3,3} = 0 + 1 + 0 = 1$ and $(V \cdot U)_{3,3} = 0 + 1 + 0 = 1$. Alternatively, for, e.g., $i = 2$ and $j = 3$, we verify that $(U \cdot V)_{2,3} = 0 + 0 + 0 = 0$ and $(V \cdot U)_{2,3} = -\frac{\lambda(1,1)}{\kappa(1,2)} - \frac{\lambda(1,1)}{\theta(1,2)} = -\lambda(1,1)/(p_2 - p_1) - \lambda(1,1)/(p_1 - p_2) = 0$.

Further, for, e.g., $i = 2$ the $i$-th row of $U \cdot D$ is $\left(1, 0, p_2, -p_3 \cdot \frac{\lambda(2,2)}{\kappa(2,3)}, p_4 \cdot \frac{\lambda(2,3)}{\kappa(2,4)}, \dots, -p_{T-1} \cdot \frac{\lambda(2,T-2)}{\kappa(2,T-1)}\right)$. Then, for, e.g., $j = 2$, we obtain $(U \cdot D \cdot V)_{2,2} = \left(1 \cdot 0 - 0 \cdot \frac{\lambda(1,1)}{\theta(1,2)} + p_2 \cdot 1\right) = p_2 = P_{2,2}$. For $j = 3$, we have $(U \cdot D \cdot V)_{2,3} = \left(1 \cdot 0 + 0 \cdot \frac{\lambda(1,2)}{\theta(1,3)} - p_2 \cdot \frac{\lambda(2,2)}{\theta(2,3)} - p_3 \cdot \frac{\lambda(2,2)}{\kappa(2,3)} \cdot 1\right) = -p_2 \cdot \frac{p_2 - 1}{p_2 - p_3} - p_3 \cdot \frac{p_2 - 1}{p_3 - p_2}$ $= (p_2 - 1) \cdot \frac{p_2 - p_3}{p_3 - p_2} = 1 - p_2 = P_{2,3}$.

All entries of the matrices $U \cdot V = 1^{T \times T}$, $V \cdot U = 1^{T \times T}$, or $U \cdot D \cdot V = \bar{P}$ can be checked similarly.

## C    Complexity of our BDS Approach (Section 3.3)

To discuss the complexity of our approach (BDS), we consider the following setup:

- $P$ is a bidiagonal matrix of size $T \times T$ (consider $A = 1$ action only).
- $K$ data observations $(x, y, n)$ from state $x$ to state $y$ within $n$ steps (with $n$ up to $N$ steps).
- the data set with $K$ observations contains a number of different combinations of $(x, y, n)$, which will be denoted by $H$.
- The cells of all $n$-step transitions matrices $P^n$ are functions of the $T - 1$ variables $p_i$, $i = 1, ..., T - 1$.
- In the likelihood objective, we require the $H$ probabilities $P^n_{x,y}$ as function of all $p_i$, $i = 1, ..., T - 1$.

**Complexity of the Standard approach (BSS)**    Required cells of various $P^k$ can be computed by recursive matrix multiplication up to $P^N$. Further, a computation of a given $P^k \cdot P$ is of $O(T^2)$ complexity as $P$ is sparse with two relevant diagonals, $k = 1, \ldots, N - 1$. Overall, to compute all $H$ required cells, i.e. $P^n_{x,y}$, we have $O(T^2 \cdot N)$ effort.

**Complexity of using Expectation Maximization (EM)**    In each iteration with certain $p_i$, $i = 1, \ldots, T - 1$, for all $(x, y, n)$ observations, we require cells of all $P^k$, $k = 0, \ldots, n$, that are characterized by the $p_i$. To compute all required cells, as in BSS, we have $O(T^2 \cdot N)$ effort (for each iteration). With multiple iterations the overall complexity for matrix multiplications is worse than BSS (but no solver is required).

**Complexity of our Explicit Diagonalization Approach (BDS)**    We have explicit formulas for $P^n = U \cdot D^n \cdot V$, where $U$, $D$, $V$ are $T \times T$ matrices, and $D$ is a diagonal matrix. To compute a required cell, i.e., $P^n_{x,y}$, we need the row $x$ of $U$ and the column $y$ of $V$. The expressions for $\lambda$, $\kappa$, and $\theta$ can be derived recursively, i.e. all with $O(T)$ effort, as follows.

First, we define auxiliary functions $f_k(z_0, z_1, ..., z_k)$, $k = 1, ..., T - 1$, with $k + 1$ arguments $z_i \in [0, 1]$, $i = 0, 1, ..., k$, using the following recursion:

$$f_k(z_0, z_1, ..., z_k) = \prod_{i=1,...,k} (z_0 - z_i) = \begin{cases} z_0 - z_1 & , k = 1 \\ f_{k-1}(z_0, z_1, ..., z_{k-1}) \cdot (z_0 - z_k) & , 1 < k < T \end{cases}.$$

Note, the effort to obtain all functions $f_k$, $k = 1, ...., T - 1$, is $O(T)$.

Next, to efficiently get the functions that are used in a row $x$ of $U$, we obtain the required $\kappa(x - 1, \cdot)$ directly via the auxiliary functions, i.e.,

$$\kappa(i, j) = \prod_{k=i,...,j-1} (p_j - p_k) = f_{j-i}(p_j, p_i, ..., p_{j-1}), \quad 1 \le i < j < T$$

and the $\lambda(x - 1, \cdot)$ functions recursively via

$$\lambda(i, j + 1) = \prod_{k=i,...,j+1} (p_k - 1) = \begin{cases} p_i - 1 & , i = j, i = 1, ..., T - 2 \\ \lambda(i, j) \cdot (p_{j+1} - 1) & , 1 \le i < j \le T - 1 \end{cases}.$$

Second, to efficiently get the functions that are used in a column $y$ of $V$, we obtain the required $\theta(\cdot, y)$ directly via the auxiliary functions, i.e.,

$$\theta(i, j) := \prod_{k=i+1,...,j} (p_i - p_k) = f_{j-i}(p_i, p_{i+1}, ..., p_j), \quad 1 \le i < j < T,$$

and the $\lambda(\cdot, y - 1)$ functions recursively via

$$\lambda(i, j) = \prod_{k=i,...,j} (p_k - 1) = \begin{cases} p_i - 1 & i + 1 = j, i = 1, ..., T - 2 \\ \lambda(i + 1, j) \cdot (p_i - 1) & , 1 \le i + 1 < j \le T - 1 \end{cases}.$$

Hence, to compute the required formulas for a row $x$ of $U$ via recursive $\lambda/\kappa$ functions has O($T$) effort. Similarly, to compute the required formulas for a column $y$ of $V$ via recursive $\lambda/\theta$ has O($T$) effort, too. Finally, the scalar product (incl. $D_{i,i}^n$), $i = 1, \ldots, T$, i.e.,

$$\bar{P}_{i,j}^{n(k)}(\vec{p}(a)) = \sum\nolimits_{t=1,\ldots,T} U_{i,t}(\vec{p}(a)) \cdot D_{t,t}^{n(k)}(\vec{p}(a)) \cdot V_{t,j}(\vec{p}(a)),$$

is also of O($T$) effort.

Overall, to compute all $H$ required cells $P_{x,y}^n$ has O($T \cdot H$) complexity. We would like to highlight, that the remarkable linear complexity in $T$ is due to the fact that we can consider single cells of $P_{x,y}^n$ matrices separately without needing to compute formulas for entire matrices of size $T \times T$. On top of that, the complexity is not affected by $n$.

**Comparison of the Complexity of BSS and BDS**  Both approaches have different complexities and they are of complementary type! If $T$ and $N$ are small and $H$ is large (i.e. close to all $0.5 \cdot T^2 \cdot N$ potential cells) then BSS should be used. If $N$ is large, particularly if $N >> T$ and $H$ is small (compared to all $0.5 \cdot T^2 \cdot N$ potential cells) then BDS is more efficient.

Recall, we consider particularly problems with large $N >> T$ and sparse data, i.e. small $K$ and $H \leq K$. Note, we seek to address the problem that standard approaches do not scale in $N$. For such problems BDS (compared to BSS) allows to reduce the linear complexity in $N$ to a constant one and to make use of setups in which $H << T \cdot N$. Note, that we look for probability estimations with sparse data observations $K$. Further, depending on the considered setup, the number $H$ of different $(x, y, n)$ can be small compared to $K$, i.e. $H << K$, especially when $K$ is large.

In sum, while the Standard approach BSS has a complexity of O($T^2 \cdot N$), our BDS approach has a complexity of O($T \cdot H$). If we use BDS to compute *full matrices* $P^n$ the complexity is O($T^3$), which still pays off if $N >> T$. Moreover, BDS requires a fraction of variables compared to BSS.

# D  EXPECTATION MAXIMIZATION (SECTION 4.2)

The EM algorithm (Dempster et al., 1977) consists of two steps - the expectation (E) and the maximization (M) step. We consider a discrete Markov chain setup.

**(0) Initialization** We (randomly) initialize starting values for $\vec{p}(a)$ for all actions $a = 1, ..., A$. Note, $\vec{p}(a)$ then characterizes all $n$-step transition matrices $\bar{P}^n(\vec{p}(a))$, $n = 0, 1, ..., N$, where $\bar{P}^0$ refers to the identity matrix.

**(1) E-Step** The E step works as follows, $i = 1, ..., T - 1$:

For all data observation $k = 1, ..., K$, we compute the expected number of 1-step transitions from $i$ to $i$ within the $n(k)$ steps (compared to all paths from $x_1(k)$ to $y(k)$) via

$$E(\Gamma_{i,i}^{(k)}) = 1/\bar{P}_{x_1(k),y(k)}^{n(k)}(\vec{p}(x_2(k)))$$
$$\sum_{t=0,1,\ldots,n(k)-1} \bar{P}_{x_1(k),i}^{t}(\vec{p}(x_2(k))) \cdot p_i(a) \cdot \bar{P}_{i,y(k)}^{n(k)-t-1}(\vec{p}(x_2(k))).$$

Here, the different combinations of transitions are organized in 3 phases: First, in $t$-steps from $x_1(k)$ to $i$, then, in 1 step from $i$ to $i$, and finally, from $i$ to $y(k)$ in $n(k) - (t+1)$ steps, $t = 0, 1, ..., n(k) - 1$.

Similarly, the expected number of 1-step transitions from $i$ to $i + 1$ is

$$E(\Gamma_{i,i+1}^{(k)}) = 1/\bar{P}_{x_1(k),y(k)}^{n(k)}(\vec{p}(x_2(k)))$$
$$\sum_{t=0,1,\ldots,n(k)-1} \bar{P}_{x_1(k),i}^{t}(\vec{p}(x_2(k))) \cdot (1 - p_i(a)) \cdot \bar{P}_{i+1,y(k)}^{n(k)-t-1}(\vec{p}(x_2(k))).$$

Here, we have $t$-steps from $x_1(k)$ to $i$, then, in 1 step from $i$ to $i + 1$, and finally, from $i + 1$ to $y(k)$ in $n(k) - (t+1)$ steps, $t = 0, 1, ..., n(k) - 1$.

**(2) M-Step** Then, the M step to update $p_i(a)$ (according to the average ratio of the likeliness of two types of paths over all data) is given by, $i = 1, ..., T - 1$:

$$p_i^{(new)}(a) = \sum_{k=1,...,K|a=x_2(k)} E(\Gamma_{i,i}^{(k)}) / \left( E(\Gamma_{i,i}^{(k)}) + E(\Gamma_{i,i+1}^{(k)}) \right).$$

Both steps (1) - (2) are repeated until convergence, where we use the threshold $\varepsilon = 10^{-6}$ for the maximum change over all $p_i$ within one iteration step.

As the fix points of the approach provide local optima, one can use different initializations to obtain different solutions. For larger $n$ the approach can become numerically unstable as $n$-step transitions may have probabilities close to 0. The approach does not require a solver. However, the computation of the entries of $n$-step transition probabilities remain costly. Further, convergence may require numerous iterations.

## E   EVALUATION TABLES (SECTION 4.3)

Table 9: Impact of the number of observations $K$; Example 2.

| $K$ | MAPE | MAE | TIME IN S |
|---|---|---|---|
| 50 | 0.143± 0.057 | 0.133± 0.054 | 0.02± 0.003 |
| 100 | 0.038± 0.015 | 0.036± 0.015 | 0.02± 0.004 |
| 200 | 0.024± 0.005 | 0.022± 0.005 | 0.04± 0.006 |
| 500 | 0.012± 0.002 | 0.011± 0.002 | 0.06± 0.005 |
| 1 000 | 0.009± 0.002 | 0.008± 0.002 | 0.10± 0.006 |
| 2 000 | 0.006± 0.001 | 0.005± 0.001 | 0.16± 0.013 |
| 5 000 | 0.004± 0.001 | 0.004± 0.001 | 0.26± 0.008 |
| 10 000 | 0.003± 0.000 | 0.003± 0.000 | 0.42± 0.020 |
| 20 000 | 0.002± 0.000 | 0.002± 0.000 | 0.75± 0.022 |
| 50 000 | 0.001± 0.000 | 0.001± 0.000 | 1.81± 0.078 |
| 100 000 | 0.001± 0.000 | 0.001± 0.000 | 3.44± 0.105 |

Table 10: Impact of the interval length parameter $G$ (in percent of $N$); Example 2.

| $G$ | MAPE | MAE | TIME IN S |
|---|---|---|---|
| 0.05 | 0.009± 0.002 | 0.008± 0.002 | 0.10± 0.005 |
| 0.1 | 0.008± 0.001 | 0.007± 0.001 | 0.10± 0.005 |
| 0.2 | 0.007± 0.002 | 0.007± 0.001 | 0.10± 0.007 |
| 0.3 | 0.008± 0.002 | 0.007± 0.002 | 0.10± 0.007 |
| 0.4 | 0.008± 0.001 | 0.007± 0.001 | 0.10± 0.007 |
| 0.5 | 0.007± 0.002 | 0.007± 0.002 | 0.10± 0.005 |
| 0.6 | 0.008± 0.004 | 0.007± 0.003 | 0.09± 0.004 |
| 0.7 | 0.007± 0.001 | 0.006± 0.001 | 0.09± 0.004 |
| 0.8 | 0.011± 0.012 | 0.010± 0.011 | 0.08± 0.004 |
| 0.9 | 0.006± 0.002 | 0.006± 0.001 | 0.06± 0.004 |

Table 11: Impact of the number of usage levels $A$; Example 2.

| $A$ | MAPE | MAE | TIME IN S |
|---|---|---|---|
| 1 | 0.008± 0.001 | 0.007± 0.001 | 0.08± 0.003 |
| 2 | 0.011± 0.002 | 0.011± 0.002 | 0.17± 0.006 |
| 3 | 0.015± 0.001 | 0.014± 0.001 | 0.26± 0.007 |
| 4 | 0.019± 0.001 | 0.018± 0.001 | 0.36± 0.007 |
| 5 | 0.021± 0.002 | 0.019± 0.002 | 0.44± 0.004 |
| 7 | 0.026± 0.001 | 0.024± 0.001 | 0.68± 0.024 |
| 10 | 0.034± 0.005 | 0.032± 0.005 | 1.04± 0.027 |

# F  EVALUATION TABLES - BELIEF DISTRIBUTION DATA (SECTION 4.4)

Table 12: Impact of the maximum number of steps $N$; Example 4.

| $N$ | MAPE | MAE | TIME IN S |
|---:|---|---|---|
| 1 | $0.061\pm 0.0040$ | $0.042\pm 0.0035$ | $0.09\pm 0.003$ |
| 5 | $0.020\pm 0.0035$ | $0.017\pm 0.0030$ | $0.13\pm 0.012$ |
| 10 | $0.012\pm 0.0013$ | $0.011\pm 0.0011$ | $0.18\pm 0.016$ |
| 20 | $0.007\pm 0.0013$ | $0.007\pm 0.0012$ | $0.28\pm 0.018$ |
| 50 | $0.004\pm 0.0005$ | $0.004\pm 0.0005$ | $0.51\pm 0.025$ |
| 100 | $0.003\pm 0.0006$ | $0.003\pm 0.0006$ | $0.80\pm 0.048$ |
| 200 | $0.002\pm 0.0002$ | $0.002\pm 0.0002$ | $1.04\pm 0.055$ |
| 500 | $0.001\pm 0.0001$ | $0.001\pm 0.0001$ | $1.35\pm 0.061$ |
| 1 000 | $0.001\pm 0.0001$ | $0.001\pm 0.0001$ | $1.87\pm 0.079$ |

Table 13: Impact of the number of observations $K$; Example 4.

| $K$ | MAPE | MAE | TIME IN S |
|---:|---|---|---|
| 50 | $0.033\pm 0.0145$ | $0.031\pm 0.0131$ | $0.06\pm 0.004$ |
| 100 | $0.020\pm 0.0070$ | $0.019\pm 0.0065$ | $0.10\pm 0.009$ |
| 200 | $0.013\pm 0.0031$ | $0.012\pm 0.0028$ | $0.15\pm 0.012$ |
| 500 | $0.010\pm 0.0019$ | $0.009\pm 0.0017$ | $0.24\pm 0.015$ |
| 1 000 | $0.008\pm 0.0030$ | $0.007\pm 0.0027$ | $0.40\pm 0.034$ |
| 2 000 | $0.007\pm 0.0011$ | $0.006\pm 0.0010$ | $0.70\pm 0.060$ |
| 5 000 | $0.006\pm 0.0006$ | $0.005\pm 0.0005$ | $1.54\pm 0.094$ |
| 10 000 | $0.005\pm 0.0004$ | $0.005\pm 0.0003$ | $2.93\pm 0.212$ |
| 20 000 | $0.005\pm 0.0002$ | $0.004\pm 0.0002$ | $5.46\pm 0.355$ |
| 50 000 | $0.005\pm 0.0002$ | $0.004\pm 0.0002$ | $13.37\pm 0.789$ |
| 100 000 | $0.005\pm 0.0002$ | $0.004\pm 0.0002$ | $26.83\pm 1.566$ |

Table 14: Impact of the number of usage levels $A$; Example 4.

| $A$ | MAPE | MAE | TIME IN S |
|---|---|---|---|
| 1 | $0.008\pm 0.002$ | $0.007\pm 0.002$ | $0.43\pm 0.048$ |
| 2 | $0.012\pm 0.014$ | $0.010\pm 0.013$ | $2.54\pm 0.193$ |
| 3 | $0.019\pm 0.025$ | $0.018\pm 0.023$ | $7.82\pm 0.561$ |
| 4 | $0.026\pm 0.034$ | $0.025\pm 0.032$ | $15.07\pm 1.837$ |
| 5 | $0.034\pm 0.041$ | $0.031\pm 0.039$ | $24.43\pm 3.557$ |
| 7 | $0.053\pm 0.045$ | $0.049\pm 0.042$ | $50.00\pm 8.565$ |
| 10 | $0.078\pm 0.040$ | $0.072\pm 0.038$ | $98.79\pm 17.57$ |

