# OpenReview forum: "Estimating Markov Chain Transition Probabilities for Steady Aging Models from $n$-step Data"
_ICLR.cc/2026/Conference — Submitted to ICLR 2026_

### Official Review · Reviewer_udKf · 2025-10-29

**Soundness:** 3
**Presentation:** 2
**Contribution:** 2
**Rating:** 4
**Confidence:** 1

**Summary:**

The paper aims to estimate aging dynamics of discrete-time Markov chains (DTMCs) from sparse n-step data, which is often intractable to optimize the likelihood function for large n.
The paper only considers a class of stead-aging processes captured by band matrices.
The paper is able to efficiently solve the likelihood through explicit diagonalization of band-structured transition matrices.
The proposed method scales well, and is applicable for large n.
The experiments on both synthetic and real data, demonstrate its high estimation accuracy.

Overall,  the targeted problems, the proposed solution and the new technical contributions are easy to follow.
The targeted problems might be only interested for a relatively small community in ICLR. Although the proposed diagonalization technique of band-structured transition matrices seems to be new, the authors fails to provide theoretical guarantees like approximation error analysis for such methods which could be more important for the others to use this approach.
Based on these concerns, I think this submissions is not ready to be published, and thus gave the weak rejection.

**Strengths:**

- Overall, the targeted problems, the proposed solution and the new technical contributions are easy to follow.

- It might be novel to consider the diagonalization technique of band-structured transition matrices for aging processes, while i am not sure how novel is such math technology for the other communities like probabilities and applied mathematics. The authors fail to clarify this in the submission.

**Weaknesses:**

- Notation clarity: The authors should improve the Notation clarity. For instance, in Eqs(1) and (3), n degree power is different!

- The studied problems might be important for industries and a small community in ICLR. I'm not sure how important the studied problems to ICLR audiences. It may be not that important for such conference.

**Questions:**

- Except the experimental results, can you provide any theoretical guarantees such as approximation error analysis, for the proposed diagonal technique of band matrices?

---

> ### Author Response · Authors · 2025-11-15
>
> Thanks for your constructive feedback and the hints how to improve the paper.
>
> (1) Notation Clarity: Sorry for the confusion. In (1) and (3) the degree is the same; in (1) the little $n$ was just placed too far right. We will take care of a clear notation.
>
> (2) Relevance: We believe that widely used standard models such as Markov Models and the estimation of their dynamics are of general interest to the data science community including ICLR.
>
> (3) Theoretical Properties: Motivated by the reviewers interest in theoretical properties of our model – in particular its complexity, we analyzed its complexity compared to other approaches.
>
> To discuss the complexity of our approach (BDS), we consider the following setup:
>
> - P is a bidiagonal matrix of size TxT (consider A=1 action only)
> - K data observations (x,y,n) from state x to y within n steps (with n up to N steps)
> - the data set with K observations contains a number H of different combinations of (x,y,n)
> - the cells of all n-step transitions matrices P^n are functions of the T-1 variables $p_i$, $i=1,…,T-1$
> - in the likelihood objective, we require the H probabilities $P^n_{x,y}$ as function of all $p_i$
>
>
> Complexity of the Standard approach (BSS)
>
> - required cells of various $P^k$ can be computed by recursive matrix multiplication up to P^N
> - a computation of a given $P^k \cdot P$ is of O($T^2$) complexity as P is sparse, $k=1,…,N-1$
> - overall, to compute all H required cells, i.e. $P^n_{x,y}$, we have O($T^2 \cdot N$) effort
>
>
> Complexity of the Expectation Maximization (EM)
>
> - in each iteration with certain $p_i$, $i=1,…,T-1$, for all (x,y,n) observations, we require cells of all $P^k$, $k=0,…,n$, that are characterized by the $p_i$
> - to compute all required cells, as in BSS, we have O($T^2 \cdot N$) effort (for each iteration)
> - with multiple iterations the overall complexity for matrix multiplications is worse than BSS (but no solver is required)
>
>
> Complexity of our Explicit Diagonalization Approach (BDS)
>
> - we have explicit formulas for $P^n=U \cdot D^n \cdot V$, where U, D, and V are TxT matrices, D diagonal
> - to compute a required cell, i.e., $P^n_{x,y}$, we need the row x of U and the column y of V
> - the expressions for lambda, kappa, and theta can be derived recursively, i.e. all with O(T) effort
> - to compute formulas for a row of U via recursive lambda/kappa has O(T) effort
> - to compute formulas for a column of V via recursive lambda/theta has O(T) effort
> - the scalar product (incl. $D^n_{i,i}$), $i=1,…,T$, see page 4 line 197, is also of O(T) effort
> - hence, overall, to compute all H required cells $P^n_{x,y}$ has O($T \cdot H$) complexity
> - The remarkable linear complexity in T is due to the fact that we can consider single cells of P^n separately without needing to compute formulas for entire matrices of size TxT.
>
>
> Comparison of the Complexity of BSS and BDS
>
> - both approaches have different complexities and they are of complementary type!
> - if T and N are small and H is large (i.e. close to all $0.5 \cdot T^2 \cdot N$ potential cells) then BSS is suitable
> - if T or N are large and H is small (compared to all $0.5 \cdot T^2 \cdot N$ potential cells) then BDS is suitable
> - note, we consider particularly problems with large N and sparse data, i.e. small K and H$\le$K
> - For such problems BDS (compared to BSS) allows to reduce the linear complexity in N to a constant one and to make use of setups in which H << $T \cdot N$. Note, that we look for probability estimations with sparse data observations K. Further, depending on the considered setup, the number H of different (x,y,n) can be small compared to K, i.e. H<<K, especially when K is large.
>
> In sum, while the Standard approach BSS has a complexity of O($T^2 \cdot N$), our BDS approach has a complexity of O($T \cdot H$) which is at most O($T \cdot K$).
>
> Thanks again for the feedback! We hope that these clarifications are helpful.

---

> > ### Comment · Reviewer_udKf · 2025-11-28
> > **Thanks for your rebuttal!**
> >
> > I'd like to thank the authors for the rebuttal!
> > Some of the feedbacks, address my concern about the notations!
> > I have looked into the other reviews.
> > As my evaluation on the topic relevance and theoretical contributions of the work does not change, I maintain my score!

---

### Official Review · Reviewer_Bswe · 2025-11-01

**Soundness:** 3
**Presentation:** 1
**Contribution:** 1
**Rating:** 0
**Confidence:** 3

**Summary:**

This work suggests the use of an explicit diagonalisation formula for banded (stochastic) matrices for efficiently calculating the likelihood of certain discrete-time Markov models.

**Strengths:**

Improving the estimation of models for battery lifetimes certainly seems important for moving away from fossil fuels.

**Weaknesses:**

**Clarity:**

I find this paper quite difficult to read due to the models not being very clearly defined. This starts with Definition 1 where I am puzzled by the terms "pre-state" and "post-state". The same issue affects Definition 2. From the later equations, I think I understand the meaning but I don't think this is standard terminology. More generally, in the introduction the manuscript would benefit from a more precise and formal descriptions of what the claimed contributions are, e.g.:
1. What precisely is the computational complexity of existing methods?
2. What precisely is the computational complexity of the newly proposed method?
3. What precisely is the novelty (is Theorem 1 novel or only its application in this context)?

And the entire class of models to be treated should be stated first before discussing the computational aspects involved in the estimation. The manuscript explains the models by stating what the shape of the observations is and then assumes that the reader infers the model from this information. This is a particular problem in the case of the "generalisation" stated on Page 5.

**Novelty:**

I am no expert in discrete-time--discrete-state Markov chain models. However, it would surprise me if the decomposition from Theorem 1 (and its subsequent use for calculating $n$-step transition probabilities) was novel. That said, from reading the paper, it is not clear to me if the authors would claim that this result is novel or merely that its use in the specific ageing-model estimation context is novel.

**Related approaches:**

There are some alternative modelling approaches such as semi-Markov models / discrete-time change-point models as well as continuous-time Markov chains. If the main claimed contribution of the present paper is the more efficient calculation of the likelihood, then It might not be necessary go discuss these. However, if the main claimed contribution is in the modelling, then it would be good if the authors could compare with such approaches.

The paper also entirely misses any discussion of hidden Markov and semi-hidden Markov models (or their continuous state or continuous time analogues such as state-space models). These are widely used in the literature on remaining-useful-life prediction including battery-life estimation. In settings in which the states are only partially or noisily observed, such models seem much more principled than the "generalisation" proposed by the authors on Page 5 (for which they also do not seem to give real-world examples).

**Questions:**

1. Since the work focuses on "bi-diagonal" transition matrices, can we not simply and cheaply evaluate the likelihood using geometric-distribution probabilities of the kind mentioned at the bottom of Page 4 (Line 215)? Can you prove that the diagonalisation-based likelihood evaluation has lower complexity?

2. Is Theorem 1 claimed to be novel?

3. Wouldn't it make more sense to fit continuous-time Markov chains here if the regime switches occur only rarely?

---

> ### Author Response · Authors · 2025-11-12
> **Theorem 1**
>
> We would like to clarify that Theorem 1 is indeed claimed as novel. We did derive this result, which is the key contribution of the paper. It would surprise us if this result already exists in the literature. In case you can point to an existing reference - please let us know! Thank you.

---

> > ### Author Response · Authors · 2025-11-15
> >
> > Thanks for the constructive feedback and the hints how to improve the paper.
> >
> > (1) Novelty of Theorem 1: Theorem 1 is claimed as novel. We should have made that clearer. It does not follow from known general results as a special case. We looked again in existing works and did not find that result.
> >
> > (2) Problem Description: Sorry for the confusion. The main problem is of course to estimate state-dependent transition probabilities in a discrete time setting from observed state transition data over certain number of time periods (n-step data). The model is standard and not claimed as contribution.
> >
> > (3) Alternative Modellings: We are aware of alternative modellings. However, the focus of the paper is not the modelling; the focus of the paper is to estimate probabilities in a discrete time setup with sparse data and large n-step intervals.
> >
> > The idea of Question 1 is nice, but geometric-distribution probabilities do not work as in our setting the probabilities are state-dependent.
> >
> > (4) Complexity of our Approach compared to Other Approaches: To discuss the complexity of our approach (BDS), we consider the following setup:
> >
> > - P is a bidiagonal matrix of size TxT (consider A=1 action only)
> > - K data observations (x,y,n) from state x to y within n steps (with n up to N steps)
> > - the data set with K observations contains a number H of different combinations of (x,y,n)
> > - the cells of all n-step transitions matrices P^n are functions of the T-1 variables $p_i$, $i=1,…,T-1$
> > - in the likelihood objective, we require the H probabilities $P^n_{x,y}$ as function of all $p_i$
> >
> >
> > Complexity of the Standard approach (BSS)
> >
> > - required cells of various $P^k$ can be computed by recursive matrix multiplication up to P^N
> > - a computation of a given $P^k \cdot P$ is of O($T^2$) complexity as P is sparse, $k=1,…,N-1$
> > - overall, to compute all H required cells, i.e. $P^n_{x,y}$, we have O($T^2 \cdot N$) effort
> >
> >
> > Complexity of the Expectation Maximization (EM)
> >
> > - in each iteration with certain $p_i$, $i=1,…,T-1$, for all (x,y,n) observations, we require cells of all $P^k$, $k=0,…,n$, that are characterized by the $p_i$
> > - to compute all required cells, as in BSS, we have O($T^2 \cdot N$) effort (for each iteration)
> > - with multiple iterations the overall complexity for matrix multiplications is worse than BSS (but no solver is required)
> >
> >
> > Complexity of our Explicit Diagonalization Approach (BDS)
> >
> > - we have explicit formulas for $P^n=U \cdot D^n \cdot V$, where U, D, and V are TxT matrices, D diagonal
> > - to compute a required cell, i.e., $P^n_{x,y}$, we need the row x of U and the column y of V
> > - the expressions for lambda, kappa, and theta can be derived recursively, i.e. all with O(T) effort
> > - to compute formulas for a row of U via recursive lambda/kappa has O(T) effort
> > - to compute formulas for a column of V via recursive lambda/theta has O(T) effort
> > - the scalar product (incl. $D^n_{i,i}$), $i=1,…,T$, see page 4 line 197, is also of O(T) effort
> > - hence, overall, to compute all H required cells $P^n_{x,y}$ has O($T \cdot H$) complexity
> > - The remarkable linear complexity in T is due to the fact that we can consider single cells of P^n separately without needing to compute formulas for entire matrices of size TxT.
> >
> >
> > Comparison of the Complexity of BSS and BDS
> >
> > - both approaches have different complexities and they are of complementary type!
> > - if T and N are small and H is large (i.e. close to all $0.5 \cdot T^2 \cdot N$ potential cells) then BSS is suitable
> > - if T or N are large and H is small (compared to all $0.5 \cdot T^2 \cdot N$ potential cells) then BDS is suitable
> > - note, we consider particularly problems with large N and sparse data, i.e. small K and H$\le$K
> > - For such problems BDS (compared to BSS) allows to reduce the linear complexity in N to a constant one and to make use of setups in which H << $T \cdot N$. Note, that we look for probability estimations with sparse data observations K. Further, depending on the considered setup, the number H of different (x,y,n) can be small compared to K, i.e. H<<K, especially when K is large.
> >
> > In sum, while the Standard approach BSS has a complexity of O($T^2 \cdot N$), our BDS approach has a complexity of O($T \cdot H$) which is at most O($T \cdot K$).
> >
> > We hope that these clarifications address your concerns.

---

### Official Review · Reviewer_yZ7N · 2025-11-03

**Soundness:** 3
**Presentation:** 2
**Contribution:** 2
**Rating:** 4
**Confidence:** 3

**Summary:**

This paper focuses on an understanding of product aging and the modelling of corresponding processes. The idea is to reconstruct the aging trajectory from a small number observations while not relying on domain expertise. The aging process itself is modelled as a Markov chain with a particular transition matrix P intended to capture the aging dynamics. To facilitate estimation of P based on observed data, the authors introduce a procedure that is based on a diagonalization of P in an explicit way. The proposed methodology is tested on several examples and is shown to perform well.

**Strengths:**

The analytical, or at least partially analytical, approach is welcomed. The resulting scalability is likely to make the method useful in at least some scenarios that are encountered in practice. The experiments are well-chosen and illustrate the results of the paper.

**Weaknesses:**

I would say that the flow of the paper is at time unclear and the main contribution is not as clearly stated as it should be for such a conference paper. The range of applications of such methodology, beyond a very specific task of estimating certain aging models, is not clear. The authors should carefully try to see where other such scenarios can be encountered.

**Questions:**

How does the model behave if the number of observations is really large? To what extent can the method be extended to other structures, beyond the band matrix?

---

> ### Author Response · Authors · 2025-11-15
>
> Thanks for the constructive feedback and the hints how to improve the paper.
>
> (1) Main Contribution: The main contribution is the explicit diagonalization result of Theorem 1. We should have made clear that this result is entirely novel! It does not follow from known general formulas as a special case. We looked again in existing works and did not find that result.
>
> (2) Range of Applications: We believe that the Markov Chain framework is of a general nature and could be used for different domains and usecases. The band matrix model for transition probabilities can be applied whenever the progress of a steady process is under consideration. Besides aging processes of in various domains, the model can be useful for life insurances, predictive maintenance, process planning, surviving models, risk assessments, etc.
>
> (3) More General Matrices: We tried to generalize the result of Theorem 1 to more general transition matrices. However, it was not possible to derive closed-form expression for more general cases. In this context, as closed-form expressions seem very rarely tractable and we are happy about having found the closed-form expression of Theorem 1.
>
> (4) Large Number of Observations K & Complexity: This is a very smart question! The number K and particularly the number (H) of different combinations of (x,y,n) in the data are responsible for the efficiency of the approach. In case K is such that basically all cells of all matrices $P^n$, $n=1,…,N$, are involved in the likelihood objective, then for all (roughly $0.5 \cdot T^2 \cdot N$) cells the underlying dependence of the T-1 variables $p_i$ has to be computed. To do that the recursive standard approach can be more efficient than our BDS approach. Nevertheless, our approach uses way less variables and hence, scales better regarding the required memory (cf. out-of-memory issues).
>
> Overall, this closely relates to the question of the complexity of our approach, which was also asked about the other reviewers.
>
> To discuss the complexity of our approach (BDS), we consider the following setup:
>
> - P is a bidiagonal matrix of size TxT (consider A=1 action only)
> - K data observations (x,y,n) from state x to y within n steps (with n up to N steps)
> - the data set with K observations contains a number H of different combinations of (x,y,n)
> - the cells of all n-step transitions matrices P^n are functions of the T-1 variables $p_i$, $i=1,…,T-1$
> - in the likelihood objective, we require the H probabilities $P^n_{x,y}$ as function of all $p_i$
>
>
> Complexity of the Standard approach (BSS)
>
> - required cells of various $P^k$ can be computed by recursive matrix multiplication up to P^N
> - a computation of a given $P^k \cdot P$ is of O($T^2$) complexity as P is sparse, $k=1,…,N-1$
> - overall, to compute all H required cells, i.e. $P^n_{x,y}$, we have O($T^2 \cdot N$) effort
>
>
> Complexity of the Expectation Maximization (EM)
>
> - in each iteration with certain $p_i$, $i=1,…,T-1$, for all (x,y,n) observations, we require cells of all $P^k$, $k=0,…,n$, that are characterized by the $p_i$
> - to compute all required cells, as in BSS, we have O($T^2 \cdot N$) effort (for each iteration)
>
>
> Complexity of our Explicit Diagonalization Approach (BDS)
>
> - we have explicit formulas for $P^n=U \cdot D^n \cdot V$, where U, D, and V are TxT matrices, D diagonal
> - to compute a required cell, i.e., $P^n_{x,y}$, we need the row x of U and the column y of V
> - the expressions for lambda, kappa, and theta can be derived recursively, i.e. all with O(T) effort
> - to compute formulas for a row of U via recursive lambda/kappa has O(T) effort
> - to compute formulas for a column of V via recursive lambda/theta has O(T) effort
> - the scalar product (incl. $D^n_{i,i}$), $i=1,…,T$, see page 4 line 197, is also of O(T) effort
> - hence, overall, to compute all H required cells $P^n_{x,y}$ has O($T \cdot H$) complexity
> - The remarkable linear complexity in T is due to the fact that we can consider single cells of P^n separately without needing to compute formulas for entire matrices of size TxT.
>
>
> Comparison of the Complexity of BSS and BDS
>
> - both approaches have different complexities and they are of complementary type!
> - if T and N are small and H is large (i.e. close to all $0.5 \cdot T^2 \cdot N$ potential cells) then BSS is suitable
> - if T or N are large and H is small (compared to all $0.5 \cdot T^2 \cdot N$ cells) then BDS is suitable
> - note, we consider particularly problems with large N and sparse data, i.e. small K and H$\le$K
> - For such problems BDS (compared to BSS) allows to reduce the linear complexity in N to a constant one and to make use of setups in which H << $T \cdot N$. Note, that we focus on problems with sparse data observations K. Further, depending on the considered setup, the number H of different (x,y,n) can be small compared to K, i.e. H<<K.
>
> In sum, while BSS has a complexity of O($T^2 \cdot N$), our BDS approach has a complexity of O($T \cdot H$) which is at most O($T \cdot K$).

---

### Official Review · Reviewer_KcWH · 2025-11-03

**Soundness:** 3
**Presentation:** 4
**Contribution:** 4
**Rating:** 8
**Confidence:** 2

**Summary:**

The paper addresses the challenge of estimating complex usage-dependent aging processes, such as battery degradation, from sparse multi-step (`n`-step) transition data using standard Markov Chain methods. These methods become computationally infeasible as `n` grows large. The authors propose a novel approach for "steady-aging" processes modeled by discrete-time Markov Chains with
band matrices (specifically bi-diagonal). The key innovation is the explicit diagonalization of the band transition matrix, enabling efficient and exact computation of multi-step probabilities (`n`-step) and allowing optimal likelihood function optimization even for large `n`.

**Strengths:**

**Efficiency gains over standard approaches**

**Evaluation uses both synthetic data (controlled environment) and real-world battery data (practical relevance).**

**High accuracy (~98% estimation accuracy).**

**Data efficiency ("comparably few data" ~10x observations per state).**

**Scalability (`n >= 1000` periods solved in seconds). This directly supports the key claim of handling large `n`.**

**Limitations and scope are clearly presented**

**Weaknesses:**

The text does not provide enough details about the exact nature and size of real-world datasets and detailed experimental setup parameters beyond `n` sizes (though Appendix A is mentioned).

The claim "near-optimal prediction results" is not explicitly quantified or compared to a theoretical optimum in the provided text.

**Questions:**

Can you be more precise about the  claim on "near-optimal prediction results" ?

---

> ### Author Response · Authors · 2025-11-15
>
> Thanks for the positive feedback and the on point summary of the paper.
>
> (1) Details of the Real-World Dataset: To test the applicability of our approach, we used real-world data of size K=3150, with T=10 and N=20 and A=1, see Table 4-5, page 7-8. We will add details to that.
>
> (2) We used the MAPE as an estimation error measure compared to the underlying true dynamics used to simulate training data. When claiming near-optimal prediction results we refer to an accuracy measure, which is defined by 1- MAPE. As the MAPE is consistently less than 2% (mostly even less than 1%), we translated that to 98\% accuracy or more.
>
> (3) We would also like to point out that we now analyzed the complexity of our approach compared to the standard one. Please see our responses to the other reviewers. While the Standard approach has a complexity of O($T^2 \cdot N$), our approach has a complexity of at most O($T \cdot K$), which shows the effectiveness of our approach in the targeted case of sparse data (small K) with long n-step transitions (large N).
>
> To discuss the complexity of our approach (BDS), we consider the following setup:
>
> - P is a bidiagonal matrix of size TxT (consider A=1 action only)
> - K data observations (x,y,n) from state x to y within n steps (with n up to N steps)
> - the data set with K observations contains a number H of different combinations of (x,y,n)
> - the cells of all n-step transitions matrices P^n are functions of the T-1 variables $p_i$, $i=1,…,T-1$
> - in the likelihood objective, we require the H probabilities $P^n_{x,y}$ as function of all $p_i$
>
>
> Complexity of the Standard approach (BSS)
>
> - required cells of various $P^k$ can be computed by recursive matrix multiplication up to P^N
> - a computation of a given $P^k \cdot P$ is of O($T^2$) complexity as P is sparse, $k=1,…,N-1$
> - overall, to compute all H required cells, i.e. $P^n_{x,y}$, we have O($T^2 \cdot N$) effort
>
>
> Complexity of the Expectation Maximization (EM)
>
> - in each iteration with certain $p_i$, $i=1,…,T-1$, for all (x,y,n) observations, we require cells of all $P^k$, $k=0,…,n$, that are characterized by the $p_i$
> - to compute all required cells, as in BSS, we have O($T^2 \cdot N$) effort (for each iteration)
> - with multiple iterations the overall complexity for matrix multiplications is worse than BSS (but no solver is required)
>
>
> Complexity of our Explicit Diagonalization Approach (BDS)
>
> - we have explicit formulas for $P^n=U \cdot D^n \cdot V$, where U, D, and V are TxT matrices, D diagonal
> - to compute a required cell, i.e., $P^n_{x,y}$, we need the row x of U and the column y of V
> - the expressions for lambda, kappa, and theta can be derived recursively, i.e. all with O(T) effort
> - to compute formulas for a row of U via recursive lambda/kappa has O(T) effort
> - to compute formulas for a column of V via recursive lambda/theta has O(T) effort
> - the scalar product (incl. $D^n_{i,i}$), $i=1,…,T$, see page 4 line 197, is also of O(T) effort
> - hence, overall, to compute all H required cells $P^n_{x,y}$ has O($T \cdot H$) complexity
> - The remarkable linear complexity in T is due to the fact that we can consider single cells of P^n separately without needing to compute formulas for entire matrices of size TxT.
>
>
> Comparison of the Complexity of BSS and BDS
>
> - both approaches have different complexities and they are of complementary type!
> - if T and N are small and H is large (i.e. close to all $0.5 \cdot T^2 \cdot N$ potential cells) then BSS is suitable
> - if T or N are large and H is small (compared to all $0.5 \cdot T^2 \cdot N$ potential cells) then BDS is suitable
> - note, we consider particularly problems with large N and sparse data, i.e. small K and H$\le$K
> - For such problems BDS (compared to BSS) allows to reduce the linear complexity in N to a constant one and to make use of setups in which H << $T \cdot N$. Note, that we look for probability estimations with sparse data observations K. Further, depending on the considered setup, the number H of different (x,y,n) can be small compared to K, i.e. H<<K, especially when K is large.
>
> In sum, while the Standard approach BSS has a complexity of O($T^2 \cdot N$), our BDS approach has a complexity of O($T \cdot H$) which is at most O($T \cdot K$).

---

### Author Response · Authors · 2025-11-20
**Revised version of the paper**

Hi all, thanks again for your time and the constructive feedback!

We included the feedback in a revised version of the paper, where we particularly (i) clarify the novelty of Theorem 1 and (ii) discuss the complexity advantages against standard approaches for the considered problem (with sparse data and large $n$-step intervals) in greater detail, see Appendix C, page 15-16.

Thanks again for your help to improve the paper!

---

> ### Author Response · Authors · 2025-11-30
> **Rebuttal Summary**
>
> The reviewer team provided constructive feedback (thanks!) and suggested three revision items, which we all addressed in the revision:
>
> (1)   (Clarification of Novelty) Reviewer yZ7N and Bswe commented that it is not clearly stated whether the results of Theorem are novel or already known. Indeed, the results ARE novel and we made that now explicitly clear in the revised paper.
>
>
> (2)   (Theoretical Results & Comparison of Complexity) Reviewer yZ7N, Bswe, and udKf asked about the complexity of the proposed solution compared to the baselines. In the revised paper, we now analyzed the complexity in full detail – with very positive results regarding the two key parameters $T$ and $N$:
>
> The O($N \cdot T^2$) complexity of standard baselines grows quadratically in the number of states $T$ and linear in $N$, the number of the largest interval length.
>
> In the revision, we were able to show that the complexity of our technique is instead O($T \cdot H$), i.e. it grows only linearly in the number of states T and is even independent of N, the number of the largest interval length.
>
> This missing analysis explains the promising numerical results obtained in the paper: As in our targeted problem the number H of different state transitions observed in the data (sparse data) is small compared to T and N (large intervals) the proposed approach clearly pays off and scales orders of magnitude better than existing solutions.
>
>
> (3)   (Details for the Real-world dataset and further potential Applications) Reviewer KcWH, yZ7N, and udKf asked for more details regarding the used NASA battery dataset and suggested to point to further applications of the model besides battery aging.
>
> We provided the missing explanations. Further, we recall that the Markov Chain framework is of a general nature and can be used for different usecases. Besides aging processes in various domains, the model can also be useful for life insurances, process planning, surviving models, risk assessment, long horizon forecasts, predictive maintenance applications and many more.
>
>
> We would like again the thank the reviewers, which helped a lot to clarify and to improve the results of the paper!

---

### Meta-Review · Area_Chair_mSjh · 2026-01-08

**Summary:**

This paper proposes a way to estimate transition probabilities in discrete Markov chains. They focus on the specific context of steady-aging processes, for which banded matrices are appropriate for modeling the transition matrix. As these can be explicitly diagonalized, the likelihood computation is shown to be very tractable and results are validated empirically.

Overall, this paper is quite narrow in scope, and my read echoes reviewer concerns in the novelty of the methodology. The written motivation switches between the motivating context of steady-aging processes, arguing that a very naive application of general Markov chain techniques would incur high computational cost, and the novelty of the proposed approach, which is limited to straightforward computation of a simplifying assumption on the transition structure that happens to be appropriate for this application. As one reviewer opines, there is a vast body of work for discrete-time, discrete-state Markov chains (which are the most prevalent class), as well as efficient algorithms developed for settings such as HMMs that the authors do not point to. Indeed, many applied papers will use some structure of their particular Markov model to speed up the computations of the likelihood; for this to meet the bar for acceptance the methodology should contain ideas that are broader in scope or have deeper theoretical novelty.

**Reviewer Concerns:**

There are concerns on the breadth of contributions and relevance to the larger ICLR community, as well as methodological novelty in the context of the existing literature on this topic as outlined in the metareview. Some reviewers acknowledge their appreciation of the author responses but it is unclear these issues are fully resolved in the rebuttal period.

**Reviewer Scores:**

Reviewer scores are consistent with their written reports and those that responded suggest that they would like to maintain the scores after rebuttal

---

### Decision · Program_Chairs · 2026-01-26

Reject